# Microfluidics: A Novel Approach for Dehydration Protein Droplets

**DOI:** 10.3390/bios11110460

**Published:** 2021-11-16

**Authors:** Van Nhat Pham, Dimitri Radajewski, Isaac Rodríguez-Ruiz, Sebastien Teychene

**Affiliations:** 1Graduate University of Science and Technology (GUST), Vietnam Academy of Science and Technology, Hanoi 10072, Vietnam; pham-van.nhat@usth.edu.vn; 2Department of Advanced Materials Science and Nanotechnology, Vietnam Academy of Science and Technology, University of Science and Technology of Hanoi (USTH), Hanoi 10072, Vietnam; 3Laboratoire de Génie Chimique, UMR 5503, 4 allée Emile Monso, 31432 Toulouse, France; dimitri.radajewski@gmail.com (D.R.); isaac.rodriguezruiz@ensiacet.fr (I.R.-R.)

**Keywords:** microfluidics, lysozyme, dehydration, equation of state

## Abstract

The equation of state of colloids plays an important role in the modelling and comprehension of industrial processes, defining the working conditions of processes such as drying, filtration, and mixing. The determination of the equation is based on the solvent equilibration, by dialysis, between the colloidal suspension and a reservoir with a known osmotic pressure. In this paper, we propose a novel microfluidic approach to determine the equation of state of a lysozyme solution. Monodispersed droplets of lysozyme were generated in the bulk of a continuous 1-decanol phase using a flow-focusing microfluidic geometry. In this multiphasic system and in the working operation conditions, the droplets can be considered to act as a permeable membrane system. A water mass transfer flow occurs by molecule continuous diffusion in the surrounding 1-decanol phase until a thermodynamic equilibrium is reached in a few seconds to minutes, in contrast with the standard osmotic pressure measurements. By changing the water saturation of the continuous phase, the equation of state of lysozyme in solution was determined through the relation of the osmotic pressure between protein molecules and the volume fraction of protein inside the droplets. The obtained equation shows good agreement with other standard approaches reported in the literature.

## 1. Introduction

The equation of state describing the interaction of colloids in solution plays a very important role in the food, paint, chemistry, and cosmetics industries, as it allows for defining the operating conditions of drying, filtration, and mixing processes. A common approach to determining this thermodynamic property of multiphasic systems is based on the equilibration by solvent exchange through a dialysis membrane, between the dispersion understudy and a reservoir with a given osmotic pressure (or water activity for the case of water). In particular, this technique has been used to study the behavior of casein [1,2,3], lysozyme [4,5], and ovalbumin [6] under compression. However, the application of this technique is hampered by the large sample volumes required for the dialysis chamber and by the permeation flux through the membranes.

Alternatively, protein solution dehydration experiments were performed by Rickard et al. [7]. In their work, the authors created single droplets of lysozyme solution with an average diameter around 50 µm using a micropipette technique and dipping the droplets in 1-decanol with a known water activity. The authors made the link between the amount of water in the continuous medium and the hydration potential of the protein. At the end of the process, glassified protein micro-beads were obtained. However, the preparation of such experiments is time consuming and the shape of the pipetted droplets is not perfect, thus potentially leading to inhomogeneous diffusion profiles.

A novel method to determine the equation of state of a protein in solution is proposed in this paper, by means of a droplet-based microfluidic approach. This methodology stands on the mass transfer between two partially miscible fluids and permits the generation of monodispersed spherical droplets of an aqueous solution of protein (lysozyme) in the bulk of a continuous phase (1-decanol) by using a high-throughput flow-focusing microfluidic configuration. As soon as the droplets are generated, water diffuses in the continuous phase, considered as an infinite reservoir, until a thermodynamic equilibrium is reached.

In this work, the influence of the experimental conditions (continuous phase water saturation fraction, initial droplet concentration, and initial droplet diameter) were studied for the dissolution/dehydration process of pure water and protein solution droplets. Droplets were continuously monitored by optical microscopy imaging, and images were automatically treated to obtain the evolution of the droplets’ volume (by measuring their diameter) as a function of time. At the end of the dissolution/dehydration process (i.e., when the equilibrium is reached), the osmotic pressure and the hydration potential of protein molecules were calculated from the final water activity value inside the droplet, which equals the known water activity in the surrounding medium. The results were discussed and compared with other works reported in the literature, and with the Carnahan–Starling equation of state.

### Theoretical Background of Droplet Dissolution

The Epstein–Plesset (E–P) model, applied to study the stability of gas bubbles in liquids since 1950 [8], shows that the evolution of gas bubble diameter is related to an under/over saturated amount of gas in the liquid. Following that model, the group of D. Needham studied the dissolution of microdroplets containing a single phase: air [9], liquid [10], or multi-phase [11] in an immiscible liquid medium. In the case of a water droplet in an organic solvent, its dissolution is only governed by diffusion, and the main variables are the concentration gradient from the droplet surface and the pressure gradient across the interface. The concentration gradient is caused by the difference between the concentration at the thermodynamic equilibrium at the droplet’s interface ($c=c_{s}$) and the concentration of the bulk solution ($c_{\infty }=c_{0}$). The dissolution of a droplet with diameter R starts intermediately after its generation. Consequently, D being the diffusion coefficient, the temporal evolution of droplet’s diameter can be calculated as $\left(dR/dt\right)=1/\rho \left(c_{0}-c_{s}\right)D\left\{1/R+1/\sqrt{\pi Dt}\right\}=-c_{s}/\rho \left(1-f\right)D\left\{1/R+1/\sqrt{\pi Dt}\right\}$, where $f=c_{0}-c_{s}$ is the saturation fraction of the surrounding medium.

## 2. Materials and Methods

### 2.1. Dehydration Experiments

Lysozyme from hen egg white (dialyzed, lyophilized, powder, ~100,000 U/mg, Sigma-Aldrich) was used as received. Ultra-pure water was used for sample preparation, filtered by a PURELAB machine (USA Resistivity of 18.18 MΩ·cm^−1^). Pure 1-decanol grade (≥99%,Sigma-Aldrich, St. Louis, MO, USA) was used and kept dry using 3A molecule sieves (Sigma-Aldrich, St. Louis, MO, USA) before use to prepare solutions with different water saturation fractions, previously defined as *f*. The saturation fraction of water in 1-decanol was prepared by mixing dehydrated 1-decanol (*f* = 0) and saturated 1-decanol (*f* = 1) in desired volume ratio.

The 1-decanol was chosen because of its water-solubility properties. The solubility of 1-decanol in water is low enough (approximately 0.01% volume) that it allows us to neglect decanol diffusion into micro-droplets with initial volumes of tens of pico-liter. On the other hand, the solubility of water into decanol is high enough to ensure that the amount of water in the droplet is much lower than the dissolution capability of the surrounding medium. For example, for a channel of 125 μm depth, 500 μm width, and a distance of 0.5 cm between two consecutive droplets, the continuous phase volume allows dissolving at least ~3500 water droplets with a ~50 μm diameter. In this way, the 1-decanol continuous phase around water droplets can be considered as an infinite medium.

In the above-mentioned experimental conditions, lysozyme solutions were explored in concentration ranges from 50 to 175 mg/mL, generating droplets of diameters varying from 60 µm to 120 µm in continuous 1-decanol phases with *f* ranging from 0 to 0.84.

### 2.2. Microfluidic System Fabrication

Standard photolithography technique was used for fabricating a master mold. Microfluidic structures were obtained subsequently by cast molding as summarized hereafter. The detailed procedure can be found elsewhere [12]. First, a rigid mold made of photoresist (Dry film WBR series, DuPont Electronic Technologies, Research Triangle Park, NC, USA) was fabricated using conventional photolithography. The dryfilm photoresist was laminated on top of a glass slide using an office laminator (A3 Mega Drive Laminator, Mega electronics, Cambridge, UK) at 100 °C. Subsequently, the photoresist was UV-exposed (λ = 365 nm) through a photomask defining the microfluidic patterns using a mask aligner (UV-KUB 2, Khloé, Saint-Mathieu-de-Tréviers, France). The final microstructures were obtained after photoresist development using a solution at 1% *w/w* of potassium carbonate (ACS reagent, 99.0%, Alfa Aesar, Karlsruhe, Germany) and magnesium sulfate (anhydrous, reagent plus, >99.5%, Sigma–Aldrich, Taufkirchen, Germany). Subsequently, a Polydimethyl Siloxane (PDMS) replica of the master mold was obtained by casting the prepolymer and curing agent mixture (Sylgard 184 elastomer Dow, Midland, MI, USA) prepared in a 10:1 volume ratio. The liquid PDMS preparation was poured into the rigid mold, degassed, and cured in an oven at 60 °C for 2 h. The elastic PDMS replica was obtained by peeling off the rigid mold and subsequently used for casting the microfluidic structure, using the UV-sensitive adhesion glue NOA-81 (NOA 81, Norland Products Inc., East Windsor, NJ, USA) as a fabrication material, as detailed below (as illustrated in Figure 1a).

Firstly, a certain amount of pre-polymer is dropped on a clean glass slide. Afterward, the PDMS mold is placed over the NOA 81. A light pressure is applied on the PDMS mold in order to remove trapped air bubbles and to remove the excess of pre-polymer. The “sandwich” system PDMS-polymer-glass slide is UV-exposed to partially reticulate and solidify the NOA 81. The PDMS mold is then carefully lifted off, releasing the polymer structures on the substrate. The complete microfluidic system is finished by closing the channel with a glass coverslip, finishing NOA 81 reticulation, and connecting inlets and outlets for solutions injection (Figure 1b). In this work, steel needles and Nanoport (Upchurch Scientific, Oak Harbor, WA, USA) were utilized. To finish the fabrication process, microfluidic channels were hydrophobized to facilitate continuous phase wetting. To this end, channels were functionalized using a fluorinated nanoparticle coating solution (Novec™ 1720 electronic grade coating, 3M™, Maplewood, MN, USA, used as received following manufacturer instructions).

The experimental setup is schematized in Figure 2a. The microfluidic chip inlets were connected to a syringe pump system (neMESYS, Cetoni GmbH, Germany) by means of Perfluoroalkoxy (PFA) tubes (Upchurch, ID250 μm; OD1/16”). The injection flow rates for the continuous and dispersed phases were optimized to control the generation of single droplets of the aqueous solution with the flow-focusing structure (Figure 2b). The droplets were released from the flow-focusing structure (50 × 50 µm section) into a contiguous larger and deeper channel (500 × 500 µm section), surrounded by the 1-decanol medium. The droplets’ evolution and movement were monitored all along their flow through the whole microfluidic channel length using an inverted Zeiss Axio Observer microscope coupled to a high-speed camera (Miro M120, Vision Research), recording at 40 frames per second and a resolution of 1200 × 800 pixels. The embedded memory of the camera allows recording movies ranging from 3 to 15 min depending on the duration of the dissolution process. Finally, the recorded movies were analyzed with an ad hoc MATLAB code to track the evolution of the droplets’ size by time.

### 2.3. Analysis of Experimental Results

#### Protein Hydration Potential

Assuming that only water inside the lysozyme solution is solubilized in the surrounding 1-decanol medium during the dissolution process, the protein amount within a droplet is constant and determined by an initially given concentration value. The water volume fraction is determined as: $\phi_{w}=1-C/\rho_{Lys}$, where C is the initial concentration of protein, and $\rho_{Lys}=1\;g/0.7\;\mathrm{mL}=1428\;g/L$ is the specific density of protein (the lysozyme specific volume which is assumed to be a constant value of 0.7 mL/g) [13]. The water in the protein solution can be found either in the protein hydration layer (in direct interaction with the protein’s surface) or as free molecules in the intermolecular (interstitial in dense packings) spaces among protein molecules. Hence, the water volume fraction can be written as $\phi_{w}=\phi_{hyd}+\phi_{int}$. Here, the $\phi_{int}$ can be theoretically calculated using a term of packing efficiency and the random jammed state of hardcore objects. It equals to 0.36 and 0.29 for spheres and ellipsoids (with an aspect ratio of 1.5), respectively [14]. However, Rickard et al. showed that the lysozyme can be packed more efficiently, with $\phi_{int}=0.07\pm 0.01$ in vacuum [7]. The authors explained that in the vacuum, the beads were completed dry (i.e., $\phi_{w}=\phi_{int}$). The experimental value of $\phi_{int}$ was calculated by the average final concentration of protein over the volume of beads.

In this work, the relation of the hydration potential and the water activity was presented. The hydration potential of lysozyme at the end of the dehydration process was expressed by the amount of hydration water:$$h=mass\;of\;water/mass\;of\;Lysozyme=\rho_{w}\phi_{hyd}/C$$
where $\;\rho_{w}$(kg/m^3^) the density of water.

At equilibrium, the chemical potential of water in the solution is equal to the chemical potential of water in the vapor phase. At low pressure (i.e., the gas phase is considered as a perfect gas), water activity is thus defined as the ratio between the partial vapor pressure of water (p) and the standard state partial vapor pressure of water at the same temperature ($p_{0}$) as:$$a_{w}=\frac{p}{p_{0}}$$

In the case of 1-decanol/water mixture, the water activity can be obtained from the saturation fraction using the following relation [15]:$$\ln\left(\frac{a_{w}}{m}\right)=\lambda_{0}+\lambda_{1}m$$
where $a_{w}$ is the water activity in the 1-decanol/water mixture, m is the molality of water in decan-1-ol (in mol/kg), and $\lambda_{0}=0.016$ and $\lambda_{1}=0.365$ are the regression coefficients.

The osmotic pressure of a colloidal solution can be derived from the water activity by the Van’t Hoff equation:(1)$$\Pi =-\frac{k_{B}T}{v_{m}}\ln\left(a_{w}\right)\;$$
where $v_{m}$ is the partial molecular volume of water (m^3^), $k_{B}$ is the Boltzman constant, and *T* the temperature (K).

Experimental evolution of the osmotic pressure as a function of the volume fraction of lysozyme inside the droplet can be used as experimental data for establishing equations of state (EOS) of protein solutions (i.e., Π vs. Φ diagram).

In the case of repulsive colloidal suspension, the osmotic pressure can be derived by applying the non-attractive rigid sphere model and the Carnahan–Starling approximation [16]. The relation between the osmotic pressure and the volume fraction of colloids is derived by the so-called Carnahan–Starling Equation of state:(2)$$\frac{\Pi }{nk_{B}T}=\frac{1+\phi_{hs}+\phi_{hs}^{2}-\phi_{hs}^{3}}{{\left(1-\phi_{hs}\right)}^{2}}$$
where $\phi_{hs}$ is the volume fraction of an hard sphere colloid, n is the number of colloids in solution and $k_{B}$ is the Boltzman constant. For lysozyme, $\phi_{hs}$ can be calculated from the gyration radius by the following expression:$$\phi_{hs}=\frac{4}{3}\pi r_{gyr}^{2}$$
where $r_{gyr}^{}$ is the gyration radius of lysozyme.

## 3. Results and Discussion

### 3.1. Dissolution Process

The use of microfluidic systems to study the dissolution of droplets has been sparsely investigated in the literature. Hence, in order to validate that the present methodology approximates ideal conditions for its study, a comparison was performed between the evolution of a pure water droplet dissolving in 1-decanol with *f* = 0.2 and the E–P modeling of a water droplet with the same initial diameter (Figure 3). It is worth noting here that the experimental data perfectly fit with the modeling until the droplet’s diameter goes below 20 μm. For a diameter smaller than 15 μm, even the observation system can no longer properly detect the droplet.

Figure 4 depicts an illustrative example of a lysozyme solution droplet dehydration process of lysozyme, in a 1-decanol medium with a water saturation fraction *f* = 0.4. As can be observed in the image sequence (Figure 4a), the droplet disappeared at a certain time (60 s) due to refractive index matching of protein solution and 1-decanol, as reported by of Rickard et al. [7]. When the chemical potentials of water in the droplet and the continuous phase are equal, i.e., when the thermodynamic equilibrium is reached, the diffusion of water stops and the droplet diameter remains constant (from 100 s to 180 s).

Figure 4b shows a comparison of the hydration experimental data of a lysozyme droplet to the E–P modeling data with the same initial diameter and saturation fraction. A deviation can be observed from the behavior expected in a pure water droplet. In this case, the dehydration process follows a three-step process, which has been depicted by dividing the diameter evolution with time into three zones. In zone 1, the shrinking of the diameter perfectly fits with the E–P model and the experimental observation for a pure water droplet. This means that the system behaves as an ideal system. The protein solution is diluted enough that the water and protein interactions are negligible. The water removed in this part can be considered as bulk water (i.e., its activity equals 1). At zone 2, when the concentration of protein inside the droplet increases, protein-water interactions increase as well, and water activity decreases, thus slowing down the droplet dissolution, as the difference in chemical potential becomes less important. In that case, as shown in Figure 4b, the droplet diameter decreases from 57 μm down to 43 μm in 50 s. Once the water activity on both sides of the droplet interface are equilibrated, the diffusion of water stops and thus the droplet diameter remains constant (zone 3).

Three different sets of experiments were carried out to study the main factors affecting the dehydration process, namely the initial droplet diameter, the water saturation fraction in 1-decanol, and the initial concentration of protein. In each set of experiments, one variable was modified while keeping the remaining factors constant. Figure 5a shows the evolution of two droplets with different diameters as a function of time. As expected, it can be clearly observed that at equal concentration and the same saturation fraction, the final droplet diameter is proportional to the initial one. Figure 5b shows the effect of water saturation in the surrounding media on the droplet’s final state. Accordingly, the droplet exposed to a lower saturation medium (*f* = 0.1) is dehydrated faster and reaches a smaller final diameter than the droplet exposed to a more saturated medium (*f* = 0.4). Finally, in the system with the same saturation fraction and the same initial diameter, the droplet with initial higher protein concentration reaches equilibrium with a larger final diameter (Figure 5c). However, in this last case, both droplets will have the same final concentration (approximately 1100 mg/mL for the experiment presented). Based on these results, it can be concluded that the final equilibrium state of the protein solution (i.e., the final volume fraction) in the droplet is only affected by the water saturation factor (i.e., the water activity) of the surrounding medium (Figure 5d).

### 3.2. The Hydration of Protein Solution

The relation between the hydration state of protein and the water activity was investigated by varying the saturation fraction from *f* = 0 to *f* = 0.84, corresponding to the water activity from 0.12 to 0.95.

It should be noted that for high values of the saturation factor, the time needed to reach the equilibrium state is longer than the residence time of the droplet in the microfluidic channel. For these experiments, the droplets were generated in the microfluidic channel and then stored in a large 1-decanol external reservoir in order to measure the final state of the droplet. In this case, only the initial final diameters were measured. The equilibrium hydration levels (*h*, mass of water per mass of protein inside the droplet) at the end of the dissolution process are shown as a function of $a_{w}$ in Figure 6, along with data published in the literature [7,17]. In all cases, a higher $a_{w}$ in the surrounding medium accordingly results in a higher water content in the final protein phase. At $a_{w}$ > 0.8, the 1-decanol dehydration data is in good agreement with the absorption isotherm and single-particle vapor sorption data. For $a_{w}$ < 0.8 our experimental data even show lower levels of hydration than the one obtained by the Rickard technique and still keep the strong agreement with the absorption isotherm. These results confirm again that our microfluidic-based dehydration technique can conduct a better approach to investigating the hydration of protein with a perfect sphere shape and near-ideal drying medium.

### 3.3. The Equation of State of The Protein

Figure 7 shows that the evolution of the osmotic pressure as a function of the volume fraction of protein inside the droplets is in good agreement with the data obtained by Rickard et al. [7] and the group of Pasquier et al. [4].

The Carnahan–Starling equation of state (Equation (2)) is also plotted in Figure 7 for lysozyme with a gyration radius of 14.64 Å [18], and it is worth noting that all the experimental data fit well with the trend given by the equation of state. In the Pasquier approach, the osmotic pressure was determined using a dialysis membrane which provides a less-dried protein state compared to the dehydration process in an organic solvent. However, in contrast to the data obtained by Pasquier et al., no crossover of the Carnahan–Starling equation is observed in our data. This behavior means that the proteins solutions may remain in a repulsive state during the dehydration process.

It should be noted that the developed experimental approach cannot be used for very extreme values of water activity. Indeed, when water activity is very high (i.e., $a_{w}\;$> 0.95) and when the concentration of water inside the continuous phase is close to the saturation concentration, the droplet dissolution time is greater than the residence time in the microfluidic chip. Thus this approach cannot capture the whole dissolution process.

On the other hand, when no water is present in the continuous phase or when $a_{w}\;$< 0.05, the transfer of water from the droplet to the continuous phase is very fast. In that case, the Peclet is larger than 1. The Peclet number is defined as:$$Pe=Jd_{0}/D$$
where *J* is the flux estimated from the loss of droplet volume per unit at the beginning of the dissolution process J=−(1/A)dV/dt and D is the protein diffusivity. At a high Peclet number, the convection dominates and the concentration gradients of the protein at the interface is more pronounced.

In that case, as the protein are amphiphilic molecules present at the decanol/water interface, the protein desorption time from the interface to the bulk is much lower than the shrinking of the interface leading to a non-negligeable concentration gradient inside the droplet. Okuzono et al. demonstrated that when the volume fraction of a colloidal suspension is close to the volume fraction of the gel formation and when the Peclet number is high, a skin layer is formed during the dehydration process [19]. In that case, the dehydration process stops before reaching the thermodynamic equilibrium (the dehydration process is stopped by the formation of a relatively nonporous membrane) and nonspherical particles are obtained (see in Figure 8).

## 4. Conclusions

In this work, a droplet-based microfluidic platform was developed to generate lysozyme solution droplets in 1-decanol media. Droplet dehydration (occurring due to the diffusion of water molecules into the unsaturated continuous phase) was monitored at various initial saturation fractions of 1-decanol, and the equation of state of lysozyme in solution was determined through the relation of the osmotic pressure between protein molecules and the volume fraction of protein inside the droplets. In addition, the influences of all factors affecting the dehydration process (i.e., initial droplet diameter, initial lysozyme concentration, and 1-decanol saturation fraction) were investigated. The results indicated that the microfluidic system was a versatile tool to provide near-ideal conditions for the dehydration process. At the end of the dehydration process (i.e., when the system reaches the thermodynamic equilibrium), the water activity inside droplet equals the given water activity in 1-decanol. Consequently, the hydration isotherms were obtained. The experimental data of hydration shown fit perfectly with the adsorption isotherms approach. It even produced a lower hydration level compared to the same dehydration by the organic solvent of other research groups. The equation of state found by this approach showed a good agreement with data reported in the literature as well as with the Carnahan–Starling approximation. Compared to other experimental techniques, the approach developed in this study allows for studying protein interactions with less amount of protein solution (few microliters of solution) and is less time consuming (less than one hour compared to a few days with dialysis experiments).

## Figures and Tables

**Figure 1 biosensors-11-00460-f001:**
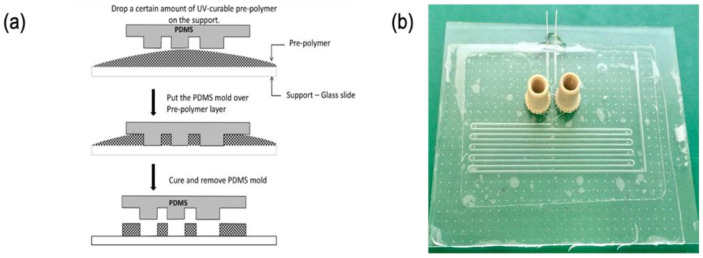
Micro-molding technique for fabrication of microfluidic systems (**a**) and the real microfluidic chip (**b**).

**Figure 2 biosensors-11-00460-f002:**
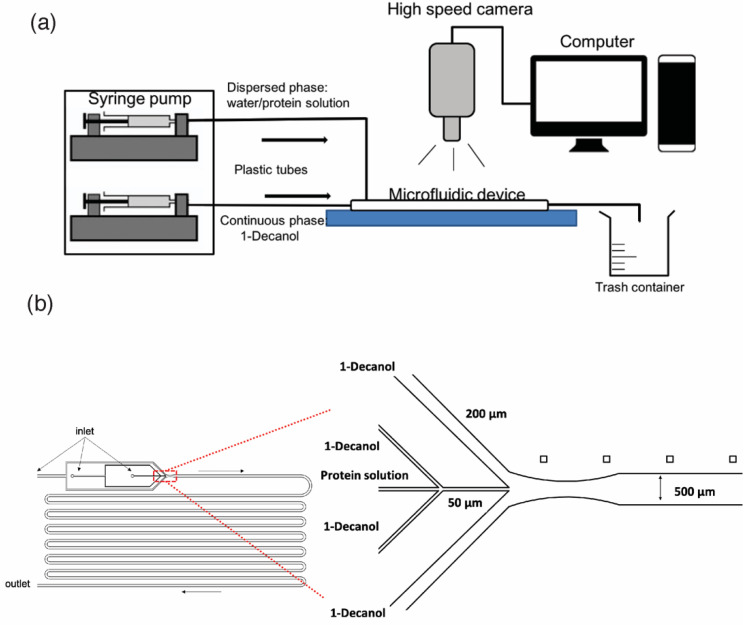
The experiment setup (**a**) and the scheme of the flow-focusing structure to generate micro-droplets (**b**).

**Figure 3 biosensors-11-00460-f003:**
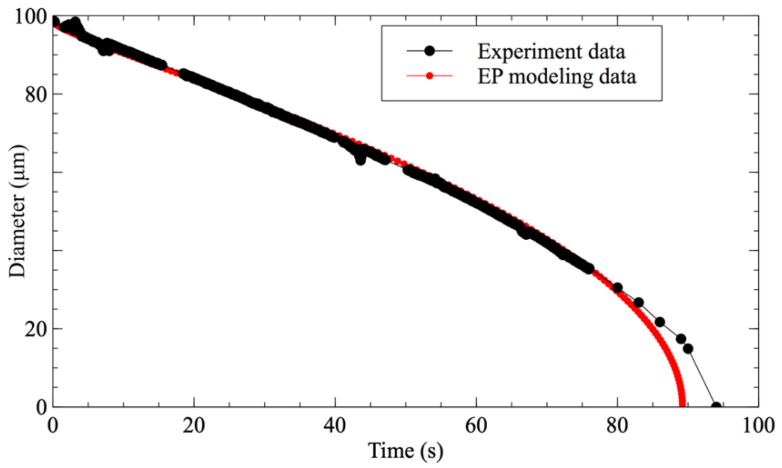
Comparison between the temporal shrinking of water droplets’ diameter in 1-decanol with saturation fraction *f* = 0.2 (black circle) and the E-P modeling data (red circle).

**Figure 4 biosensors-11-00460-f004:**
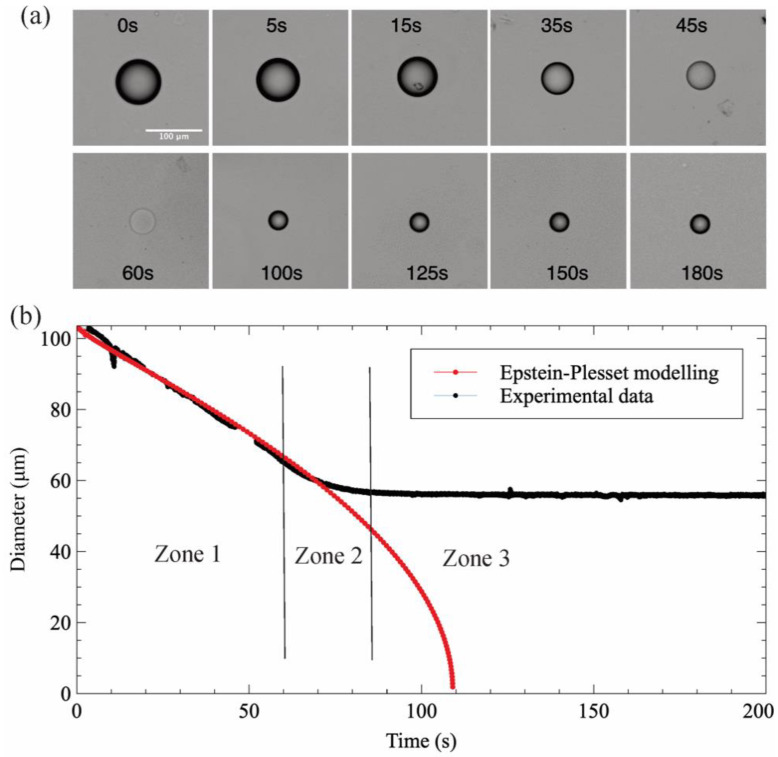
The temporal evolution of a lysozyme droplet in 1-decanol with saturation fraction *f* = 0.4. (**a**) the captured images of dehydration and (**b**) the comparison with the E–P modeling data.

**Figure 5 biosensors-11-00460-f005:**
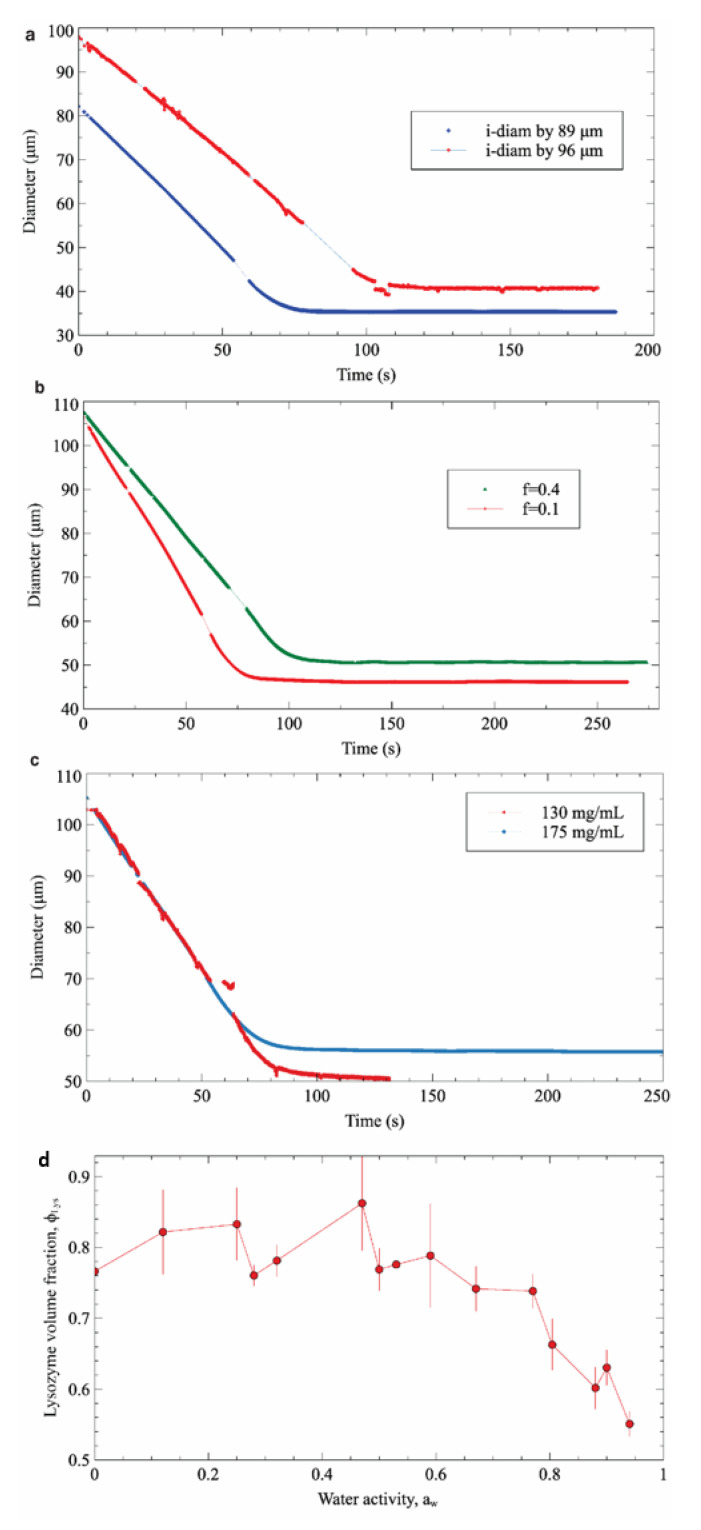
The effect of 3 factors: diameter (**a**), the saturation fraction (**b**),the initial concentration of protein (**c**) to the dehydration process, and (**d**) the final volume fraction of lysozyme as function of water activity.

**Figure 6 biosensors-11-00460-f006:**
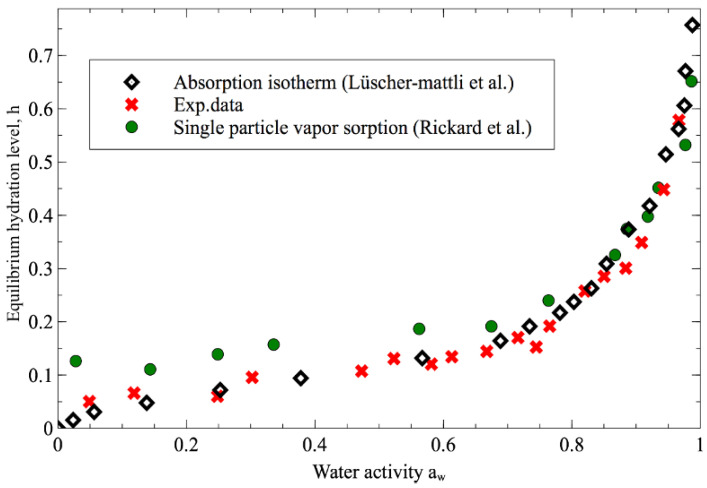
Equilibrium hydration levels at the end of the protein droplets dehydration process using 1-decanol are shown as a function of water activity to other previously published data (data of absorption isotherm obtained from ref [17] and data of single particle vapor sorption from ref. [7]).

**Figure 7 biosensors-11-00460-f007:**
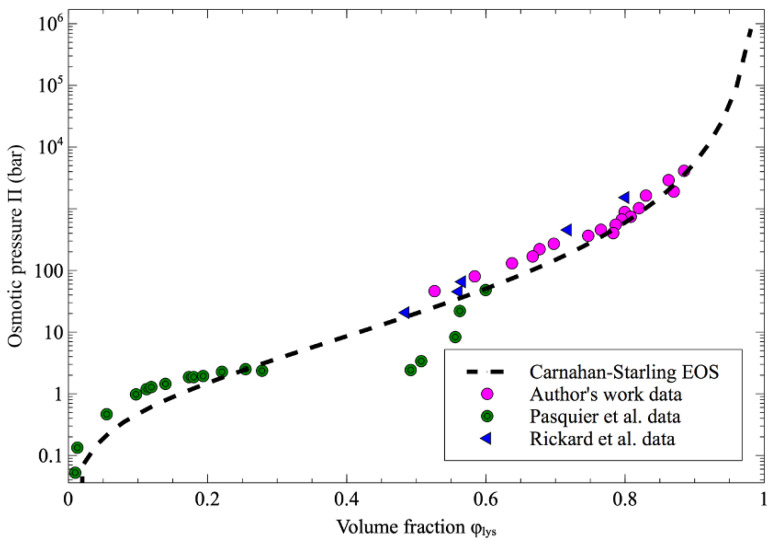
Osmotic pressure as a function of volume fraction of lysozyme. The author’s work data from this work (pink-filled circle) are compared with osmotic pressure values previously reported in the literature (green-bullseyes for data from Pasquier et al. [4], blue-filled triangle for data from Rickard [7] and the dash line for Carnahan–Starling approximation).

**Figure 8 biosensors-11-00460-f008:**
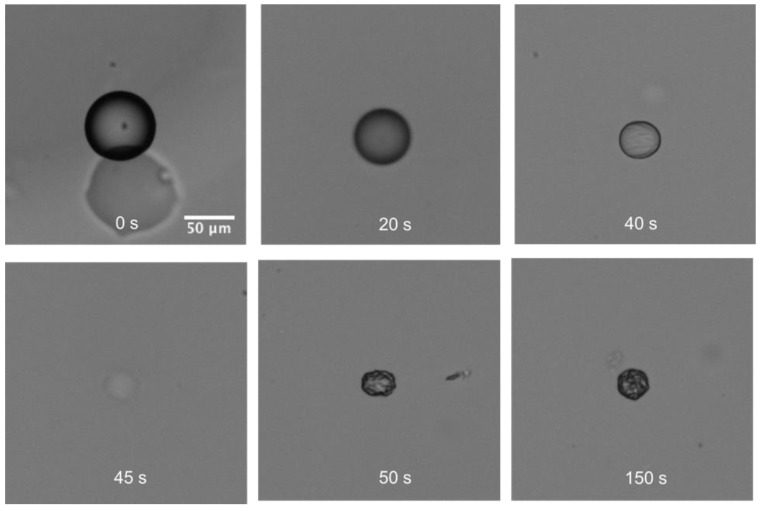
The formation of skin layer of protein during dehydration process with *f* = 0 (i.e., $a_{w}\;$ < 0.05).

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
