# Peer review of "Microfluidics: A Novel Approach for Dehydration Protein Droplets"

_biosensors, 2021, doi:10.3390/bios11110460_

Round 1

Reviewer 1 Report

In this manuscript, the authors present a droplet-based microfluidic system to investigate the protein dehydration equation. The results and the reviewed equations have shown pretty good matchup, and the paper is written in decent manner with detailed method descriptions. The only suggestion here is that the data plotted in the Figure 3 -7 should be illustrated with the supporting statistics, which should include the standard deviations and the numbers of repeat for each data point. I would recommend this manuscript to be published after minor revision.

Reviewer 2 Report

1. The authors should provide the definition for water activity a_w at the point where it is introduced in the text.

2. Figure 2b is too small and thus not readable.

3. In lines 145-146 you claim 50 x 50 um section is contiguous to 500 x 500 um section. How to you fabricate a channel with two depths, 50 um and 500 um in a single step?

4. Line 158: Where did you get the equation for the water volume fraction? The numerical constant part (/1000) in any equation is always interesting… Wouldn’t be smarter to maybe just use the unit mL/mg when providing the protein specific volume? Personally, I would prefer the equation phi_w=1-C/rho_p instead, where rho_p = 1g/0.7mL = 1428 g/L is the specific density of the protein.

5. Line 175: ‘The hydration potential of Lysozyme h = ????h? ?? ?????⁄????h? ?? ???????? = 1000?? ?h??/?, where ?? (g/m3)’. Weight is the force exerted on a body by gravity. You should use ‘mass’! You can get rid of the numerical constant by using SI units for density, kg/m^3.

6. ‘Alternatively, the water activities were converted from the saturation fraction as shown in the works of Segatin and Klofutar [15].’ Can you show the equation for this conversion in your manuscript?

7. Line 184: please provide the value of v_w, the molecular volume of water, that you use in your calculation of the osmotic pressure.

9. What are the parameters of the EP model that fit the experimental data of Figure 3?

10. Figure 4b is too small, completely unreadable. It should be at least 2x wider.

11. Figure 5 and text in l. 245: “Three different sets of experiments were carried out to order to study the main factors affecting the dehydration process…” In Fig 5 for each factor only two different values are shown. Could you add a third curve, e.g. a droplet with the initial diameter of 60 um in (a), the saturation factor of 0.8 in (b) and the initial protein concentration of 50 mg/ml in (c)?

12. Figure 5 is way too small. In the printed version the text is unreadable.

13. When I zoom in Fig5 (c), I see the initial protein concentration of 175 mg/ml whereas in the text you claim ‘in concentration ranges from 50 to 150mg/mL, generating droplets of diameters varying’ in line 101. Which of the two statements is incorrect? 

14. Line 251: You claim ‘Figure 5.b, shows the effect of water saturation in the surrounding media on droplets final state. Accordingly, the droplet exposed to a lower saturation medium (f=0) is dehydrated faster and reaching a smaller final diameter than the droplet exposed to a more saturated medium (f =0.4). ‘ The reality: In Fig 5b the droplet exposed to f=0 dehydrates SLOWER than that in f=0.4, the rate is at least 2x slower. Why?

15. Figure 6: the y-axis label is ridiculous (h=g water/g lysozyme). Why not simply put “Equilibrium hydration level”? This also applies to l. 274: ‘h, grams of water per gram of protein inside the droplet’, use mass of water per mass of protein or simply water/protein mass ratio.

16. Figure 7, caption: (unfilled circle) . I don’t see any unfilled circles in the chart. In the chart legend put “experimental data” instead “Author’s work data”.

Reviewer 3 Report

The work by Pham et al. presents an interesting study on the dehydration of protein droplets using microfluidics. The work is well-driven and the results are sound. However, authors should address a couple of issue before publication:

-The meaning of the concept of equation of state is not clear in the manuscript. What are the authors trying to explain? This aspect should be better explained.

-What is the role of the interface in the process? Lysozyme present surfacta activity at water/alkane interfaces, and it can be adsorbed at the droplet/continuous phase interface.

Round 2

Reviewer 2 Report

Thank you for the implementation of the proposed corrections.

Reviewer 3 Report

The authors have addressed all my concernings, and now the article is fully publishable.